# The influence of turbulent bursting on sediment resuspension under unidirectional currents

Sarik Salim[1], Charitha Pattiaratchi[1], Rafael Tinoco[2], Giovanni Coco[3], Yasha Hetzel[1], Sarath Wijeratne[1], Ravindra Jayaratne[4]

[1]School of Civil Environmental and Mining Engineering and UWA Oceans Institute, University of Western Australia, 35 Stirling Highway, Crawley, WA 6009, Australia
[2]Department of Civil and Environmental Engineering, University of Illinois at Urbana-Champaign, Urbana, IL 61801, USA
[3]Faculty of Science, University of Auckland, Auckland 1142, New Zealand
[4]School of Architecture, Computing and Engineering, University of East London, Docklands Campus, 4-6 University Way, London E16 2RD, UK

*Correspondence to*: Sarik Salim (sarik.salim@research.uwa.edu.au)

**Abstract.** Laboratory experiments were conducted in an open channel flume with a flat sandy bed to examine the role of turbulence on sediment resuspension. An acoustic Doppler velocimeter (ADV) was used to measure the instantaneous three-dimensional velocity components and acoustic backscatter as a proxy to suspended sediment concentration. Estimates of sediment transport assume that there is a mean critical velocity that needs to be exceeded before sediment transport is initiated. This approach does not consider the turbulence flow field that may initiate sediment resuspension through event based processes such as the 'bursting' phenomenon. In this paper, laboratory measurements were used to examine the sediment resuspension processes below and above the mean critical velocity. The results within a range above and below the measured mean critical velocity suggested that: (1) the contribution of turbulent bursting events remained identical in both experimental conditions; (2) ejection and sweep events contributed more to the total sediment flux than up-acceleration and down-deceleration events; and, (3) wavelet transform revealed a correlation between the momentum and sediment flux in both test conditions. Such similarities in conditions above and below the measured mean critical velocity highlight the need to re-evaluate the accuracy of a single time-averaged mean critical velocity for the initiation of sediment entrainment.

## 1 Introduction

Understanding the physical processes that govern sediment resuspension has significant implications for aquatic ecosystems and fish habitats as well as sustainable engineering applications such as beach nourishment, maintenance of hydraulic structures, dam breaching flows, sedimentation in reservoirs, defence schemes against erosion due to floods, and aggregate dredging (Buffington, 1999; Paphitis, 2001; van Rijn et al., 2007; Thompson et al., 2011; Aagaard and Jensen, 2013; van Rijn, 2013), all of which require improved predictive models of sediment transport. However, resuspension of sediment is a complex mechanism due to the difficulty in defining the fluctuating nature of turbulent flow. Shields (1936), the pioneer to investigate the entrainment of granular particles, concluded that a mean critical or threshold shear stress existed below which particles did

not move. At velocities lower than the threshold, shear stress represented the viscous drag imparted by the moving fluid to the bed particles whereas at velocities higher than the critical, it was related to the pressure differential between the upstream and downstream sides of the particle. Shields also defined the non-dimensional critical shear stress, $\theta_{cr}$, as a function of the boundary Reynolds number, $Re_p$, defined as:

$$\theta_{cr} = \tau_o/(\rho_s - \rho)gd_s \qquad\qquad (1)$$
$$Re_p = u_* d_s / \upsilon \qquad\qquad (2)$$

where, $\tau_o$ is the critical bottom shear velocity, $\rho_s$ and $\rho$ are the sediment and fluid densities, $g$ is the acceleration due to gravity,

$d_s$ is the particle diameter, $u_* = \sqrt{\tau_o/\rho}$ is the critical shear velocity and $\upsilon$ is the kinematic viscosity of the fluid.

Such criterion (commonly used via a Shields diagram, e.g., Kennedy, 1995; Buffington, 1999; Paphitis, 2001) states that sediment is entrained once bed shear stress exceeds the Shields mean critical value. The Shields diagram has been extensively applied and investigated by numerous researchers (Brownlie, 1981; van Rijn, 1984; Pattiaratchi and Collins, 1985;

Soulsby and Whitehouse, 1997; Wu and Wang, 1999; Paphitis, 2001). The impact of turbulence, however, was traditionally represented only by a mean quantity such as Reynolds shear stress (e.g. widely used bedload and suspended load formulations presented in van Rijn, 2013). Further attempts to characterise sediment entrainment advocated that it solely depended on fluid lifting force, with near bed sediment being entrained due to instantaneous near bed vertical velocity (Einstein, 1950; Velikanov, 1955; Yalin, 1963; Ling, 1995). In contrast, Bagnold (1956) hypothesised that particles remain in suspension as long as the

turbulent eddies have dominant vertical velocity components, which would scale with the flow shear velocity, that exceed the particle settling velocity. It implies that to establish a dynamic equilibrium of sediment exchange, the flow must continuously pick up the sediment at the same rate with an upward velocity equalling terminal fall velocity.

The critical bed shear stress concept asserts that bedload grain does not move below the mean critical value of bed

shear stress. However, Lavelle and Mofjeld (1987) studied historical data for incipient sediment motion and found that no true threshold value existed, and bedload transport could occur at any predicted threshold. This suggested that a single critical shear stress should not be included as an essential parameter when calculating bedload transport rates, agreeing with previous work from Paintal (1971) who observed that there was no distinct shear stress below which no single grain entrained. Laursen et al. (1999) found that many values of the critical shear stress could be found for an equal-sized sediment particle, matching a

similar number of sediment transport formulas available at the time. Since earlier developed diagrams showed a gap within the smooth and rough-flow regimes (Yalin and Karahan, 1979), further attempts conducting additional experiments and analysing the problem theoretically based on deterministic and probabilistic approaches, have been made to amend the Shields diagram to account for turbulent effects. Greater details on this approaches can be found in the comprehensive surveys made

by Miller et al. (1977), Buffington and Montgomery (1997), Paphitis (2001), and, Dey and Papanicolaou (2008). Conclusions reached by these authors agree that a single mean value of shear stress is not an accurate estimate for sediment transport, and further consideration must be given to instantaneous turbulent parameters for a better characterisation of flow-sediment interactions.

## 1.1 Turbulent bursting

Kline et al. (1967) found a cyclic process with turbulent flow near walls, in which the near-wall layer propagated slowly and then interacted strongly with the outer layer—an event known as 'turbulent bursting'. At the beginning, the low-speed streak ejected away from the wall, and oscillations in both the spanwise and normal directions appeared. As the oscillations increased in amplitude, a breakdown (burst) occurred in the form of a violent and chaotic upward eruption of the low-speed fluid in the near-wall layer into the outer layer, termed usually as ejection. The ejection was soon followed by a sweep, in which the chaotic motion was swept away. The wall-layer streaks reappeared at different spanwise locations, and a new quiescent period began. The development of a horseshoe vortex showing the lifts, stretches, ejection, and sweep associated with velocity profiles is shown in Fig. 1. The action of turbulent coherent flow structures related to such a sequence of turbulent bursting involving ejections and sweeps (Robinson, 1991) has been shown to play a central role in sediment entrainment (Cao et al., 1996).

This discovery of the turbulent bursting phenomenon led researchers to study the role of turbulence on particle entrainment and re-define criteria of sediment motion (Dey, 2011). Several laboratory studies have linked coherent motions in the turbulent boundary layer with resuspension (Grass, 1974; Sumer and Oguz, 1978; Sumer and Deigaard, 1981, Falco, 1991). Grass (1974) filmed the resuspension process due to turbulent flow over a flat sand bed, identified the coherent flow structures in the boundary layer, and calculated the velocities of the particles advected by such motions. This directly led to the conclusive link between the observed ejection of fluid away from the boundary layer and the corresponding response of bed sediment. Their work also showed that the sweep events above the channel bed were more responsible for momentum transfer into the boundary layer than the ejection events. Sumer and Oguz (1978) and Sumer and Deigaard (1981) photographed intermittent, sweep-type fluid motions pushing sediment particles into the low-speed wall streaks; those particles were then subjected to upward, ejection-type fluid motions. Falco (1991) formulated an overall picture of the structure of the turbulent boundary layer in terms of experimentally identifying inner-outer wall region multiscale turbulent eddies and constructed a coherent motion model. Considering a flat plate zero pressure gradient boundary layer, this study showed that a specific set of coherent structures in the turbulent boundary layer were dynamically significant for the transport of sediments. Further studies (Kaftori et al., 1995; Nelson et al., 1995; Niño and Garcia, 1996; Cellino and Lemmin, 2004) confirmed the importance of the bursting events in sediment resuspension and transport in fluvial environments. Previous studies suggested that the ejections were associated with entrainment of sediment particles into the water column, while sweeps were effective at transporting bedload (Cao, 1997; Dyer and Soulsby, 1988; Heathershaw, 1979; Soulsby, 1983; Keylock, 2007; Yuan et al., 2009). To

distinguish between different processes, in this study the term 'resuspension' is used for particles initially laying on the bed and at some point lifted into the water column, in contrast to particles permanently in suspension (i.e., washload).

Heathershaw and Thorne (1985) conducted experiments in tidal channels flowing over sandy gravels in order to study the role of turbulent structures on sediment entrainment, and showed that entrainment was correlated with the near wall instantaneous streamwise velocity, and not with the instantaneous Reynolds shear stress. Drake et al. (1988) studied gravel mobility in alluvial streams and found that most of the gravel entrainment was associated with sweep events, which occurred during a small fraction of time at any particular location of the bed. The entrainment process was thus found to be episodic: short periods of high entrainment were interspersed with long periods of weak or no entrainment. Thorne et al. (1989) observed that turbulent coherent structures were the main transporters of coarse sedimentary material. Their experiment suggested that an instantaneous increase in streamwise velocity fluctuations generated excess boundary shear stresses, which drove the transport. Soulsby et al. (1994) made simultaneous measurements of the high frequency fluctuations of concentration of sand suspended by a tidal current, and the horizontal and vertical components of the water velocity above the sandy bed of an estuary, and found that the large, upward sediment fluxes in the boundary layer were associated with ejection events. Kularatne and Pattiaratchi (2008) performed field experiment in the wave-induced flow environment of Floreat Beach, Perth, Western Australia, and concluded that higher sediment movements are associated with ejections rather than sweeps. In the tidal current environment of western Yellow Sea of China, Yuan et al. (2009) conducted experiments and noticed that ejection and sweep events caused most of the observed turbulent sediment flux.

Seminal work of Grass (1970) and Lavelle and Mofjeld (1987), along with the above-mentioned laboratory and field investigations have called to revise the critical velocity concept, proposing alternative statistical views of particle motion. Adrian (2007) investigated the structure of near bed organised motion in the canonical forms of wall turbulence and suggested that quadrant analysis permitted evaluation of the turbulent bursting events to the total mean values of kinetic energy and dissipation. Diplas et al. (2008) performed laboratory experiments to examine the role of turbulent fluctuations on particle movement under incipient flow conditions, and concluded that the duration of instantaneous turbulent events applied on a sediment grain was also significant in determining the sediment grain's threshold of motion. In an attempt to propose a direct numerical simulation of bed load transport calculations, Schmeeckle and Nelson (2003) developed a model of bed load transport that captured the sources of fluid turbulence variability by directly integrating the equations of motion of each particle of a simulated mixed grain-size sediment bed. However, they also mentioned that with the knowledge of the velocity structure within the bedload layer, a complete model of bedload transport could be built that includes the importance of turbulence fluctuations in entraining grains at low to moderate transport stages, and also includes the feedback that moving grains have on the fluid velocity in the whole bedload layer, which is important for moderate to high transport stages. The entrainment of coarse sediment particles under the action of fluctuating hydrodynamic forces was investigated from an energy perspective by Valyrakis et al. (2013). They found that the energy approach to grain dislodgement, although directly linked to the impulse

criterion, demonstrated to be more versatile and intuitive, where the majority of the turbulent events performed sufficient mechanical work on the coarse grain for entrainment. Therefore, while research that moves beyond Reynolds stresses to incorporate quadrant analysis and ejection-sweep processes is an important advance (Dwivedi et al., 2011; Wu and Shih, 2012), further attempts can be taken to link two dimensional quadrants and three dimensional octants into sequences that reveal flow-sediment structure (Keylock et al., 2014).

Despite several attempts to develop a precise sediment entrainment theory merging turbulence features, it is widely recognised (e.g., Dey, 2011) that the effect of turbulent coherent structures on sediment motion and resuspension is yet to be fully understood. The aim of the paper, rather than developing a better transport equation, is to highlight the importance of instantaneous events on sediment resuspension, which were not considered when using the classical Shields diagram approach that uses a mean velocity concept. While the stochastic characteristic of turbulence discussed by Grass (1970) and posterior observations by Lavelle and Mofjeld (1987) demonstrated the need for using statistical tools to better conceptualise the process of sediment motion, our approach takes a step further by (a) assessing the risk of over estimation of widely used sediment transport predictors (e.g. Shields, 1936; van Rijn, 1984; Soulsby, 1997; Soulsby and Whitehouse, 1997) following a mean critical velocity approach, and (b) verifying the relevance of such mean critical velocity concepts in terms of turbulent bursting phenomena. In this regard, we performed laboratory experiments where high frequency acoustic data were recorded in fluvial conditions near the bottom boundary layer under unidirectional currents over a flat sandy bed. Data collected were post-processed using Reynolds decomposition, quadrant analysis, and wavelet transform methods, to clarify the turbulent characteristics and their effect on resuspension, both above and below the measured mean critical velocity test conditions.

## 2 Methodology

### 2.1 Laboratory set-up and experimental conditions

The experiments were conducted in a 54-m-long, 2-m-wide current flume located at the University of Cantabria, Santander, Spain. The flume contained an 18-m-long, 0.20-m-deep, purpose-built sand bed (Fig. 2). The sediment was well-sorted silica sand with a grain size of $d_{50}$ = 0.31 mm with water depth, D = 0.16 m and 0.42 m.

The three-dimensional, instantaneous flow velocities were measured using two Nortek Vectrino acoustic Doppler velocimeter (ADVs) with a sampling frequency of 50 Hz. The ADVs were located above the sand bed at distances of 5.5 m (ADV 1) and 8.5 m (ADV 2) from the beginning of the sand bed (Fig. 2 at an elevation, z=5 cm above the bed). Data from the near-bed ADV 1 is presented in this manuscript where the mean flow speeds, ū, varied from 0.087 to 0.256 m/s, covering a range of boundary Reynolds number, $Re_p$={342-1004}; flow Reynolds number, $Re_D$=(ūD/υ)={$1.4 \times 10^4$-$4.1 \times 10^4$} and Rouse number, P= $w_s/ku_*$= {2.89-8.14} where $u_*$ was calculated using the bed shear stress computed with Eq.4 at z=5 cm, ū was mean velocity, $k_s$ was the von Kármán constant (assuming as 0.41) and $w_s$ was particle fall velocity calculated from Dietrich (1982).

The physical dimensions of the instruments determined the distance above the bed such that the sensor did not touch the flume bottom and would not be buried in the sand during the experiments. Since no bedforms developed during the experiments, the height of the sensors was constant for each test. The sand was flattened manually with a floor squeegee before each series (see Tinoco and Coco, 2014, 2016, for more details about the experimental set-up).

### 2.2 Data analysis techniques

Three experiments, each lasting five minutes, were conducted to study the effect of turbulent bursting on the resuspension of sediment in the range of above the measured critical velocity (AMCV) and below the measured critical velocity (BMCV) test runs. The critical resuspension velocity ($\bar{u}_{cr, measured}$ = 0.163 m/s) was obtained through data from Optical Backscatter Sensors (OBS) located at the same height of the ADVs. The threshold was considered when OBS started recording the concentration higher than the background meaning critical velocity was taken as the point of shifting the 'mean' concentration from one point to the higher point (Tinoco and Coco, 2014, 2016). The $\bar{u}/\bar{u}_{cr, measured}$ ratio for AMCV was between 1.04 and 1.57, and for BMCV was between 0.53 and 0.94. The results from two time series ($\bar{u}/\bar{u}_{cr,measured}$= 1.23 AMCV and $\bar{u}/\bar{u}_{cr,measured}$= 0.59 BMCV) were chosen for detailed analysis in order to compare above and below the time-averaged measured critical velocity conditions. For both runs, we used data from the ADV 1 located 5 cm above the flat sand bed and 5.5 m from the upstream edge. The measured mean critical velocity was 0.163 m/s and the measured water depth was 0.16 m. Two time series (both from AMCV and BMCV runs) from three experiments at this depth were also used for comparison in the quadrant analysis results, and results from a two-minute segment of those two cases are shown for better clarity. The remaining three

experiments with D=0.42 m and z=5 cm indicated similar trends, with bursting events occurring below and above the expected measured mean critical values.

Voulgaris and Trowbridge (1998) showed that ADVs can accurately measure mean flows, Reynolds stresses, and vertical turbulent components close to the bed within one percent of the estimated true values. Time series records of the ADVs' high frequency (50 Hz) velocity components (where u = horizontal flow velocity, v = transverse flow velocity, and w = vertical flow velocity) were analysed using Reynolds decomposition (Fox et al., 2004), such that the flow was assumed to be composed of mean (overbar) and fluctuating (prime) parts:

$$u = \bar{u} + u', \qquad v = \bar{v} + v', \qquad w = \bar{w} + w'. \tag{3}$$

For easier visualisation, a one-second mean of the 50 Hz velocity time series was used. To comprehend the characteristics of the bursting events, the conditional statistics of the velocity fluctuations (u' and w') were plotted into the quadrants of a u'-w' plane (Lu and Willmarth, 1973), where u' is the turbulent velocity's horizontal component and w' is the vertical component. Quadrants were named as ejection (u'<0, w'>0), sweep (u'>0, w'<0), up-acceleration (u'>0, w'>0), and down-deceleration (u'<0, w'<0) ( Heathershaw and Thorne, 1985; Kularatne and Pattiaratchi, 2008; Thorne, 2014; Schmeeckle, 2015). Work from Keylock et al. (2014), has suggested the use of extending quadrant analysis into three dimensions (known as octant analysis) characterising dominant flow structures, which can be linked to the entrainment of sediment from the bed and into suspension, and whose frequencies would dominate the velocity spectra and contribute the majority of the total shear stress. However, widely used two dimensional quadrant approach involving u'-w' plane, was chosen for this manuscript due to the simplicity of its implementation and its efficacy in revealing aspects of turbulent flow physics that otherwise have remained unexplored.

Turbulent kinetic energy (TKE) shear stress was estimated using the three components of turbulent velocity (u', v', and w') near the bed (at z=5 cm):

$$\tau_{TKE} = 0.5\rho C_1(u'^2 + v'^2 + w'^2), \tag{4}$$

where $\tau_{TKE}$ is the TKE shear stress, $\rho$ is the fluid density, and $C_1$ is a coefficient, which can be taken as 0.19 or 0.2 (Kim et al., 2000; Biron et al., 2004). In this analysis, $C_1$=0.19 was used to calculate the TKE shear stress.

The turbulent Reynolds stress was estimated near the bed as (Fox et al., 2004; Thorne, 2014):

$$\tau_{Re} = -\rho(u'w'). \tag{5}$$

ADVs backscatter was used as a representation of suspended sediment concentration (SSC) based on the following equation (Fugate and Friedrichs, 2002; Voulgaris and Meyers, 2004):

$$EL = 0.43Amp + 20log_{10}(R) + 2\alpha_w R + 20R \int \alpha_p dr, \qquad (6)$$

where EL is the echo level in dB, Amp is the amplitude in counts recorded by the ADV, R=0.05 is the range or distance between the transducer and focal point in meter, $\alpha_w$=0.6 (when salinity = 0 ppt for 1.5 MHz frequency, chosen from list of values provided in Lohrmann, 2001) is the water absorption in dBm$^{-1}$, and $\alpha_p$ is the particle attenuation in dBm$^{-1}$ (Lohrmann, 2001). At low concentrations, the particle attenuation becomes very small (Lohrmann, 2001), therefore the fourth term (i.e. $20R \int \alpha_w dr$) was ignored in this study. Additionally, to better interpret the backscatter reading as a proxy of SSC, the signal processing digital 'Butterworth' filter was used as described in Thomson and Emery (2014). Since higher SSC produces higher backscatter amplitudes, EL is used to identify instantaneous increases of SSC resulting from sweeps and ejections. We used a concentration proxy ($c'$) as an indicator to identify variations in concentration of sediment in suspension which was also analysed using Reynolds decomposition (Fox et al., 2004), where the concentration proxy was assumed to be composed of mean (overbar) and fluctuating (prime) parts:

$$c' = EL - \overline{EL} \qquad (7)$$

Wavelet analysis was used to identify localised variations of power within the time series (Torrence and Compo, 1998). The recorded time series were decomposed into time-frame space, and the dominant modes of variability and their variation in time were analysed as described in Grinsted et al. (2004). To limit the edge effects, the time series represented the region of spectrum where the effects might have been important (near large scales) by a 'Cone Of Influence (COI)' following Torrence and Compo (1998). Farge (1992) suggested that Continuous Wavelet Transform (CWT) unfolds the dynamics of coherent structures and measures their contribution to energy spectrum. Therefore, CWT was employed to derive the time evolution of momentum and sediment flux of turbulent coherent structures near the bottom boundary layer. Wavelet Coherence (WTC) was also applied in order to expose regions with high common power showing phase relationships between the CWT of momentum and sediment flux.

## 2.3 Calculation of the threshold velocity

The mean velocity threshold for sediment movement was calculated using an average grain diameter ($d_{50}$) of 0.31 mm, a grain density ($\rho_s$) for wet sand of 1905 kgm$^{-3}$, g is the gravity and freshwater density at room temperature of 1000 kgm$^{-3}$. Assuming the von Kármán constant as 0.41 and Nikuradse's roughness $z_0$ was estimated using:

$$z_0 = \frac{k_s}{30}\left(1 - exp\left[\frac{-u_* k_s}{27\upsilon}\right]\right) + \frac{\upsilon}{9u_*}$$

with $k_s = 2.5 d_{50}$

Several critical values can be thus calculated, ranging from 0.21-0.32 m/s, as shown in Table 1.

5    Table 1. Theoretical mean critical values for sediment entrainment at z=0.05 m compared in this study.

| Criteria | Equations | Calculated $\bar{u}_{cr}$ (m/s) |
|---|---|---|
| Shields (1936) | $\bar{u}_{cr} = \frac{u_*}{k} \ln\left(\frac{z}{z_0}\right)$ <br><br> $\bar{u}_* = \sqrt{[\theta_{cr}(s-1)gd_{50}]}$ <br><br> $\theta_{cr} = 0.24 D_*^{-1}$ (for $1 < D_* \leq 4$ condition), <br><br> $D_* = d_{50}\left[\frac{(s-1)g}{\upsilon^2}\right]^{\frac{1}{3}}$ | 0.210 |
| van Rijn (1984) | $\bar{u}_{cr} = 0.19\, d_{50}^{0.1}\, log_{10}\left(\frac{4D}{d_{90}}\right); 100 < d_{50} < 500\ \mu m$ <br><br><br> $\bar{u}_{cr} = 8.5\, d_{50}^{0.6}\, log_{10}\left(\frac{4D}{d_{90}}\right); 500 < d_{50} < 2000\ \mu m$ | 0.297 |
| Soulsby (1997) | $\bar{u}_{cr} = 7\left(\frac{D}{d_{50}}\right)^{1/7}[g(s_p - 1)d_{50}f(D_*)]^{1/2}$ <br><br> $s_p = \frac{\rho_s}{\rho} = \frac{density\ of\ the\ sediment}{density\ of\ the\ fluid}$ <br><br> $D_* = \left[\frac{g(s_p-1)}{\upsilon^2}\right]^{1/3} d_{50}$ <br><br> $f(D_*) = \frac{0.30}{1+1.2D_*} + 0.055(1 - e^{-0.020D_*})$ for values of $D_* > 0.1$. | 0.259 |
| Soulsby and Whitehouse (Soulsby, 1997) | $\bar{u}_{cr} = \frac{u_*}{k} \ln\left(\frac{z}{z_o}\right)$ <br><br> $u_* = \left(\frac{\tau_{cr}}{\rho}\right)^{1/2}$ <br><br> $\tau_{cr} = \theta_{cr}g(\rho_s - \rho)d_{50}$ <br><br> $\theta_{cr} = \frac{0.30}{1 + 1.2D_*} + 0.055(1 - e^{-0.020D_*})$ | 0.312 |

**3 Results**

The scatterplots of the Reynolds and TKE bottom shear stresses for the AMCV and BMCV runs (Figs. 3a and 3b) showed that higher bed shear stress (i.e. values >5 N/m$^2$ of TKE and Re shear stress estimations of both AMCV and BMCV runs) was produced to generate sediment resuspension (as evidenced with backscatter intensity on Figs. 4c and 5c). Such comparison of
the TKE and Re shear stress methods also suggested the presence of coherent flow structures in the turbulent flow which created highly localised and persistent variability near the bed, hence affecting the bed shear stress.

The velocity fluctuations (u′, w′), Reynolds shear stress (u′w′) and backscatter over a two-minute period (for better visualisation of bursting events) from the AMCV and BMCV runs were compared identifying ejection and sweep events (Fig.
4, 5, respectively). This comparison offered considerable insight into the contribution of turbulence in terms of the events associated with sediment resuspension. Overall, in the time series significant variability and intermittency both in Reynolds stress (u′w′) and sediment resuspension (backscatter) was also revealed. Such intermittent nature of u′w′ was expected and observed previously in the laboratory (Grass, 1974; Sumer and Oguz, 1978; Sumer and Deigaard, 1981; Niño et al., 2003; Schmeeckle, 2015) and in the field (Heathershaw and Thorne, 1985; Drake et al., 1988; Soulsby et al., 1994; Kularatne and
Pattiaratchi, 2008 and Yuan et al., 2009). In more detail, the time series of the AMCV run showed twenty-eight major resuspension events (Fig. 4). Eighteen of these events demonstrated ejections (at 5, 9, 17, 24, 30, 38, 49, 54, 66, 77, 83, 86, 98, 99, 101, 107, 109 and 116s) and ten of these events revealed sweeps (at 21, 25, 32, 42, 46, 53, 58, 61, 75 and 90s), which confirmed that high resuspension events were mostly associated with ejection and sweep type motions than up-acceleration and down-deceleration events during the analysed record. The same pattern was observed for the two-minute period of BMCV
run where twenty-five major resuspension events were observed (Fig. 5). Fifteen of these events were identified as ejections (at 2, 7, 19, 26, 38, 47, 52, 72, 77, 87, 90, 93, 100, 113 and 116s) and ten of these events confirmed sweeps (at 1, 32, 41, 46, 54, 60, 67, 79, 107 and 112s). Such resuspension events identified below the measured critical velocity support the theory of the non-existence of a unique time-averaged critical shear stress as suggested by Paintal (1971) and Lavelle and Mofjeld (1987). The plot of the BMCV run further indicated that though flow conditions were below the critical velocity conditions;
sediment resuspension was observed due to ejection and sweep events.

Contributions to u′w′ were also observed in four quadrants of the u′-w′ plane with a threshold value (backscatter above 10 dB) both for AMCV and BMCV runs (Figs. 6). The plots clearly showed that the large contribution of u′ and w′ were associated with ejections and sweeps rather than up-acceleration and down-deceleration events. AMCV results were similar
with previous studies (Cellino and Lemmin, 2004; Yuan et al., 2009). The distribution of turbulent components for BMCV in the u′-w′ plane reflected a similar pattern which established that resuspension events can occur even below a critical threshold value. BMCV conditions, where mean velocity was 59% of the critical velocity, showed a similar behavior to AMCV

conditions. Similarities were also found in the other data sets within the range of $\overline{u}/\overline{u}_{cr, measured}$ ratio; for AMCV between 1.04 and 1.57, and for BMCV between 0.53 and 0.94.

We performed a quadrant analysis to determine the frequency of different bursting events and their contributions to the Reynolds stress (i.e. u′w′). The occurrence percentages of four types of bursting motions, as well as their contributions to the momentum flux (u′w′) and sediment flux (c′w′) for the AMCV and BMCV experiments, are shown in Figs. 7 and 8, respectively. The results for the u′w′ signals for the AMCV and BMCV experiments agreed with the results from earlier studies (Wallace et al., 1972; Willmarth and Lu, 1972). For both AMCV and BMCV experiments, ejection and sweep events were the dominant source of the Reynolds stress; however, although the time occupied by ejection was comparable with, or even less than, that of sweep, ejection contributed more to the net Reynolds stress (AMCV = 49%; BMCV = 43%) as shown in Figs. 7a,b and 8a,b. Ejection (AMCV = 38%; BMCV = 38%) and sweep (AMCV = 37%, BMCV = 30%) mainly generated the upward sediment flux (Figs. 7c and 8c), which suggested the intense upwelling of low-speed fluid parcels with high sediment entrainment events was the main source of the overall sediment flux. In contrast, up-acceleration (AMCV = 12%; BMCV = 14%) and down-deceleration (AMCV = 13%; BMCV = 18%) events transported less sediment (Figs. 7c and 8c). Thus ejection and sweep contributed more to the total turbulent sediment flux (AMCV = 75%; BMCV = 68%) than up-acceleration and down-deceleration events (AMCV = 25%; BMCV = 32%). Such consistent results in both AMCV and BMCV confirm the need to develop transport rate formulas that consider instantaneous Reynolds stress concepts along time-averaged critical velocities.

Continuous Wavelet Transforms (CWT) and Wavelet Coherence (WTC) analysis (Grinsted et al., 2004) for AMCV and BMCV runs offered a more intuitive way to visualise the turbulence data in both time and space (Figs. 9 and 10, respectively). In the presented scalograms, at higher periods (i.e. low frequency events), the power felt within the range of COI (i.e. the shaded region in the scalograms) which limited the capability to investigate the temporal evolution of the specific peak frequencies as stated in Section 2.2. Hence, investigation was restricted to examine high frequency events occurring at time scales up to 32s for both runs. Overall, the scalograms (Figs. 9 and 10) traced the dynamics of coherent structures and its measured contribution to the sediment flux. It also revealed that within the large-scale motions (considering period bands >0.5s as large scale motions), there existed multi-scale [e.g. in AMCV time series between ~47-52s, period band ranging ~2-8s (large scale) and ~0.0625-1s (small scale); in BMCV time series between ~82-85s, period band ranging ~2-8s (large scale) and ~0.0625-2s (small scale)] and some embedding small fine-scale (e.g. in AMCV at ~22-25s, period band ranging ~0.0625-0.5s; in BMCV at ~22-23s, period band ranging ~0.0625-1s) features. This suggested that both for AMCV and BMCV runs, near the bed, most of the energy was concentrated within the high period (warmer colour >0.5s) associated with the mean flow properties for both momentum flux and sediment flux. Results also showed that highly energetic turbulent events (i.e. warmer colour >0.5s) occurred:

i) sporadically throughout the time series (e.g. in AMCV at 5, 9, 17, 21, 24 etc; in BMCV at 1, 2, 7 19 etc), especially in gradually developing clusters (considering clusters developed taking >2s time) that sustained short periods (i.e. lasted <1s) in the dominant streamwise-vertical plane of the flow near the bed,

ii) for longer periods (up to several seconds from a turbulence perception, in our case ~2-10s), vertically in the water column, and

iii) at lower frequencies for both runs. The larger clusters felt over ~1 and 8s period band for both AMCV and BMCV runs; while the fast evolving clusters (considering those lasting up to 2s) stretched between ~0.0625 and 0.5s period band before weakening.

This was evident in the colour coded contours (Fig. 9a) which were associated with ejection and sweep events for AMCV runs (Table 2a). Similarly, for BMCV runs; it was evident with ejection and sweep events (Table 2b). In addition to that, in AMCV runs; momentum flux corresponded to the contour in sediment flux within similar period bands both in ejection and sweep events as shown in Figs. 9a and b in relation to Table 2a. Similar pattern was also observed in BMCV runs in the ejection events, as well as in the sweep events where momentum and sediment flux coincide with each other showing similar period bands (Figs. 10a and b, in relation to Table 2b). The WTC was applied to the momentum and sediment flux for both runs where common features were noticed as shown in Figs. 9c and 10c in relation to Table 2a and 2b. Both for AMCV and BMCV runs, during the identified ejection and sweep events (as mentioned in Table 2a, 2b) the coherence were found to be higher (i.e. warmer colour >0.5s), suggesting that the transport mechanism greatly relies on the production of momentum flux by coherent structures in order to contribute to the sediment flux. For instance, the ejection event identified at 9s in the AMCV run (Table 2a, Figure 9c) shows higher correlation between momentum and sediment flux (i.e. warmer colour > 0.5s) with period band ranging between ~0.5s and 3s. Similar trend was observed throughout the time series of AMCV and BMCV runs.

Table 2. Major ejection (shaded cells) and sweep (white cells) events in the presented AMCV and BMCV time series.

| Condition | Time (s) | | | | | | | | | | | | | | | | | | | | | | | | | | | |
|---|---|---|---|---|---|---|---|---|---|---|---|---|---|---|---|---|---|---|---|---|---|---|---|---|---|---|---|---|
| (a) AMCV | 5 | 9 | 17 | 21 | 24 | 25 | 30 | 32 | 38 | 42 | 46 | 49 | 53 | 54 | 58 | 61 | 66 | 75 | 77 | 83 | 86 | 90 | 98 | 99 | 101 | 107 | 109 | 116 |
| (b) BMCV | 1 | 2 | 7 | 19 | 26 | 32 | 38 | 41 | 46 | 47 | 52 | 54 | 60 | 67 | 72 | 77 | 79 | 87 | 90 | 93 | 100 | 107 | 112 | 113 | 116 | - | - | - |

## 4 Discussion

In this study, the well-known Shields criterion, estimated using mean velocities, along with some of the most commonly used empirical curves (i.e. van Rijn, 1984; Soulsby 1997 and Soulsby and Whitehouse 1997, which are also derivatives of Shields diagram) were investigated in order to re-examine the prediction of sediment threshold performance (Fig. 11). In the figure, the grey shaded areas defined the range of the AMCV and BMCV mean velocities presented in this study. The calculated critical values using different approaches were shown in red dotted lines. Our measured critical velocity is clearly below the calculated Shields (1936); van Rijn (1984); Soulsby (1997) and, Soulsby and Whitehouse (1997) critical velocity conditions [i.e. measured mean critical velocity, $\overline{u}_{cr, measured}$=0.163 m/s < 0.210 m/s (Fig. 11d), 0.259 m/s (Fig. 11e), 0.297 (Fig. 11f) and 0.312 (Fig. 11g) respectively]. This suggested that the widely used above-mentioned empirical methods which are believed to be significant for the design of movable-bed channels as well as for future experimental investigations, potentially overestimated the transport of sediment by 1.05, 1.28, 1.49 and 1.56 times considering Shields (1936); van Rijn, 1984; Soulsby 1997 and Soulsby and Whitehouse 1997 (Fig. 11) approaches respectively. Both reported cases, with mean velocities of AMCV ($\overline{u}$= 0.200 m/s) and BMCV ($\overline{u}$=0.096 m/s), above and below our measured threshold ($\overline{u}_{cr, measured}$=0.163 m/s), showed evidence of sediment in suspension, further showing that the mean critical stress approach also underpredicts the transport of sediment. Although it is still common to conceptualise the mechanics of sediment transport as a time-averaged approach, this approach sustained due to the lack of enough experimental and/or field data to perform stochastic analyses. Availability of such data, as those we present, advance understanding of the turbulence structure and their role in transport processes.

Comparison of test results where mean velocity was 1.23 times higher as well as 0.59 times lower than the measured mean critical velocity showed strong similarities without major exceptions (Figs. 4 and 5). Although near bed velocity and average transport rate were greater in AMCV runs, the peak instantaneous Reynolds stress were close (i.e. $u'w' > 0.05$ m$^2$/s$^2$ in the identified peak ejection and sweep events shown in the Figs. 4b. 5b) in both AMCV and BMCV runs. Both ejection and sweep events contributed to the forward momentum flux as well as sediment flux, showing that the concept of time-averaged critical velocity by itself cannot provide a full representation of the physical processes in action in the resuspension of sediment.

In both tests (AMCV and BMCV), ejection and sweep events were the largest contributors to momentum transfer. Up-acceleration and down-deceleration events leaded to marginal effect on transport of momentum and sediment flux compared to the other two events (Fig. 6). Previously, performing quadrant analysis Heathershaw and Thorne (1985), Nelson et al., (1995) and performing octant analysis, Keylock et al. (2014) advised that up-acceleration and down-deceleration events were the individually effective means of resuspending sediments, however less net sediment flux was accomplished by these events in our AMCV and BMCV runs. These could be related to the strength of the up-acceleration and down-deceleration events which were much weaker and could not carry sediment particles to a higher level where the sampling volume was

placed (i.e., 5 cm above the bed). It is also noteworthy to mention that up-acceleration and down-deceleration events contributed less significantly with a positive stress.

Buffington and Montgomery (1997) put forward a survey suggesting that many attempts have so far been made to modify the Shields diagram, conducting additional experiments and analysing the problem theoretically based on deterministic and probabilistic approaches. Several researchers have presented laboratory or field evidence supporting the close correlation between the instantaneous sediment flux and instantaneous streamwise velocity (u), suggesting that only sweeps and up-accelerations play a significant role in the entrainment and transport of sediment, since these motions were associated with positive u′ and thus greater streamwise velocities (Thorne et al., 1989; Nelson et al., 1995; Weaver and Wiggs, 2008). However, our investigation in the AMCV and BMCV conditions showed similarities with other research groups which documented that sweeps and ejections were the primary contributors to sediment entrainment (Grass, 1970, 1974; Sumer and Deigaard, 1981; Best, 1992; Niño and Garcia, 1996; Hurther and Lemmin, 2003). In contrast, direct numerical simulations (DNS) provide a new tool for examining turbulent structure of the flow (e.g. Mathis et al. 2013). However, further development is required to apply the DNS approach to intermittent turbulent bursting events both in fluvial and geophysical flows (Venditti et al., 2013). For example, Mathis et al. (2014) estimated bed shear-stress using conventional methods and DNS modelling approach and reported a large disparity between the two methods where an order of magnitude difference between the levels of energy spectra observed. While DNS has the potential to develop methodologies for the prediction of bursting events and associated sediment resuspension mechanisms, its application on large-scale, complex flows still remains limited (e.g. Schmeeckle and Nelson, 2003). Experimental investigations such as the one developed herein, will allow to use new and existing data from acoustic velocimetry sensors to further identify and characterise such turbulent events. Direct observations of bursting events will in turn better inform DNS methods to better account for flow interactions.

Quadrant analysis showed that, in BMCV runs, ejection (in which low speed fluid moves away from the boundary towards the outer layer) entrained particles away from the bed in order to maintain them in suspension as it was in AMCV runs (Figs. 7 and 8). Sweeps, (in which high-speed fluid moves near the wall) with a negative contribution, impacted on the particles in resuspension by pushing them towards the bed. Moreover, the time occupied in both AMCV and BMCV runs were almost identical and contributed in similar percentage to instantaneous momentum and sediment flux as well. Diplas et al. (2008) demonstrated that in addition to the magnitude of the instantaneous turbulent forces applied on a sediment grain, the duration of these turbulent forces is also important in determining the sediment grain's threshold of motion, and that their product, or impulse, is better suited for specifying such conditions. This was evident in our results both in AMCV and BMCV conditions where the time occupied by the ejection and sweep events (which were also evidenced to play the dominant role in the momentum flux and sediment flux) were significantly higher in comparison to the up-acceleration and down-deceleration events. The understanding of accounting temporal contribution of bursting events presented in this study as well as discussed in Diplas et al. (2008) and Diplas and Dancey (2013) calls for consideration of the hydrodynamic impulse (i.e. value of force

multiplied by required time for the accomplishment of the event) as a comprehensive criterion in the development of future models to predict particle entrainment.

Wavelet analysis was useful to diagnose characteristics of turbulence in order to explain information about the spatial structure of the flow. Particularly, we were interested in its frequency content and energy variation (Figs. 9 and 10). Previously, experimental investigation by Shugar et al., (2010) showed that stacked series of wavelet plots indicated clusters of low-frequency coherent flow structures initiated close to the bed, grew with height above the bed and then broke-up as they were advected downstream, with their decay possibly being linked to topographically-induced flow acceleration. The frequency at which these structures were generated was suitably predicted by the models of Driver et al. (1987) and Simpson (1989) for variation in separation zone size and wake flapping, respectively. Our measured data in BMCV runs were consistent with AMCV runs as well as with previous investigations. Therefore, it can be stated that the cross wavelet transform method was effective at visualising and detecting the coherent structures from the raw turbulent data, which enabled us to study the correlation between wall turbulence structures and sediment resuspension.

## 5 Conclusions

This manuscript reports on an investigation on the validity of using the mean critical shear velocity of sediment to define thresholds of sediment resuspension. Although Lavelle and Mofjeld (1987) previously reviewed the concept of critical stress for the initial motion of non-cohesive sediment beds under turbulent flow conditions suggesting the non-existence of true threshold in the movement of sediment, their conclusions were based on photographic observations employed in conjunction with current measurements to infer sediment thresholds in the field. Likewise, the work from Niño and Garcia (1996), Niño et al. (2003) identified such instantaneous events from high-speed videos, which limit the number of captured and analysed events. We examined the influence of turbulent coherent structures on sediment resuspension for flows both above and below the measured mean critical resuspension velocity over a flat sandy bed using widely used acoustic instruments. The presented methodology can be used on existing data sets from researchers using ADVs or ADCPs in either laboratory or field settings to identify turbulent structures and their effect on suspended sediment concentration if synchronous records of acoustic backscatter exist. Such observations presented in this paper are also necessary to clarify our view of turbulent coherent structures in resuspending sediments both in low and high Reynolds-number flows while leading widespread application of DNS.

Our results show that the measured mean critical velocity alone is not sufficient to predict episodic initiation of motion, as turbulent events can move sediment even at mean flow conditions below the thresholds defined by time-averaged stresses. Measured fluctuations of turbulent Reynolds stress evidenced to move sediments at lower turbulent stresses than expected. Instantaneous particle entrainment occurred earlier than the suggested measured time-averaged critical velocity due

to the stochastic nature of turbulence. Although nearbed shear stress can be used to estimate bedload transport, significant special variations in the magnitudes and durations of the ejection, sweep, up-acceleration and down-deceleration play a significant role in sediment resuspension. The implications of sediment motion at Reynolds shear stress below the expected critical conditions further suggested that instantaneous shear stress has an important contribution to entrain particles, which

cannot be predicted with a time averaged critical velocity.

To the best of our knowledge, there is no universal agreement on identifying a unique threshold for initiation of motion or resuspension of sediment (e.g., how many grains rolling, for how long, over what area coverage) in the literature. Our study shows that turbulent bursting events produce sediment resuspension even at mean velocities well below such typical

critical values. Our statistical assessment suggests that the existing definition of threshold can be improved by incorporating turbulent effects for a more accurate description of the processes involved which will result in better predictions of sediment transport. The results of this study are instrumental in resolving an important research question: how best to incorporate the turbulent bursting events into a theoretical model describing the sediment entrainment process? The analysis detailed herein on identification of bursting events and their contribution toward the near-bed Reynolds shear stress production governing

sediment motion provide new avenues to answer such question, incorporating the use of wavelet analysis on time series of acoustic backscatter or signal intensity readily available from commonly used acoustic velocimetry instruments (ADVs and ADCPs) as a powerful tool for investigating such processes. The fact that a similar methodology can be applied to existing field and laboratory datasets that focused on velocity but collected an indicator of signal backscatter as part of the data record, further highlights its potential in future research to elucidate a more complete understanding of the interactions between flow

and sediment transport over complex topography.

**Acknowledgement**

This work was a part of the first author's PhD research when he was in receipt of Scholarship for International Research Fees (SIRF), University International Stipend (UIS) and Safety Net Top-Up Scholarship awarded by the UWA Oceans Institute and the School of Civil Environmental and Mining Engineering at the University of Western Australia, Australia. Authors

acknowledge the support of the University of Cantabria, Spain. GC funded by the Natural Hazards Research Platform (C05X0907). In addition to that, authors would like to express their appreciation to Dr Florence Verspecht and Ruth Gongora-Mesas for their assistance in preparing the final manuscript.

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

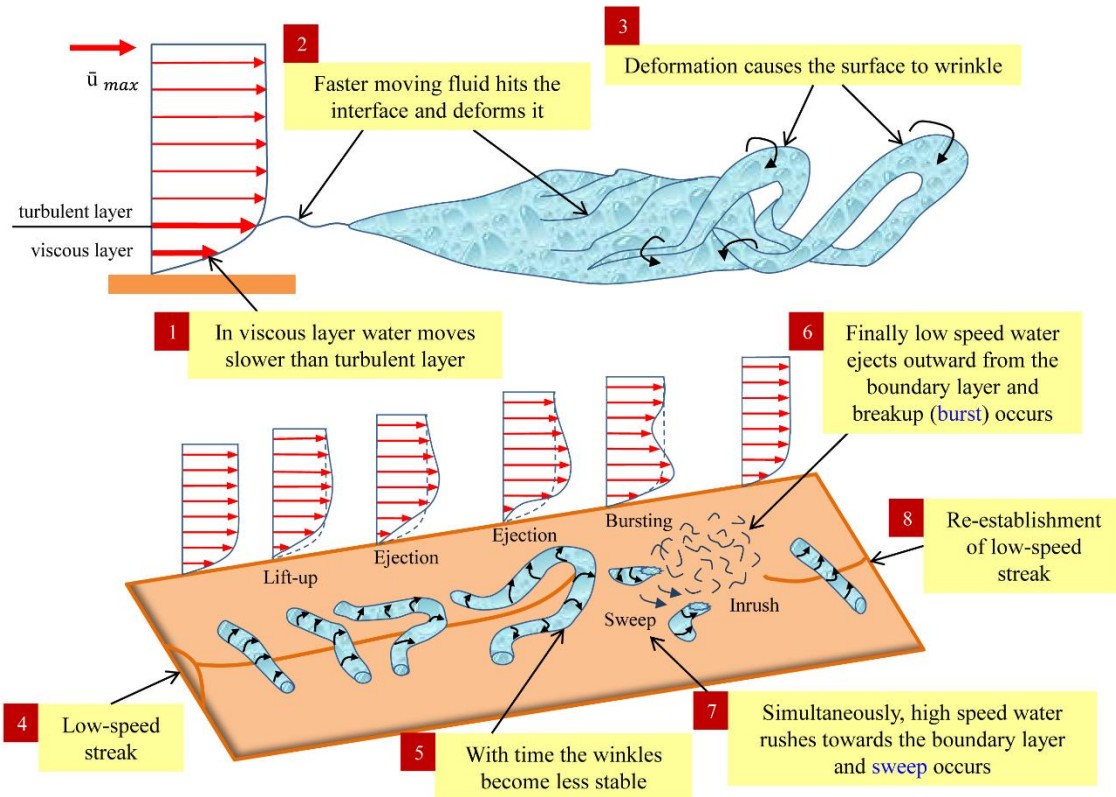

**Figure 1. Schematic diagram of the typical sequence of turbulent bursting phenomena (Allen, 1985; Robinson, 1991; Bridge, 2003) where the flow is directed from left to right and the arrow length represents the relative velocity in the velocity profiles.**

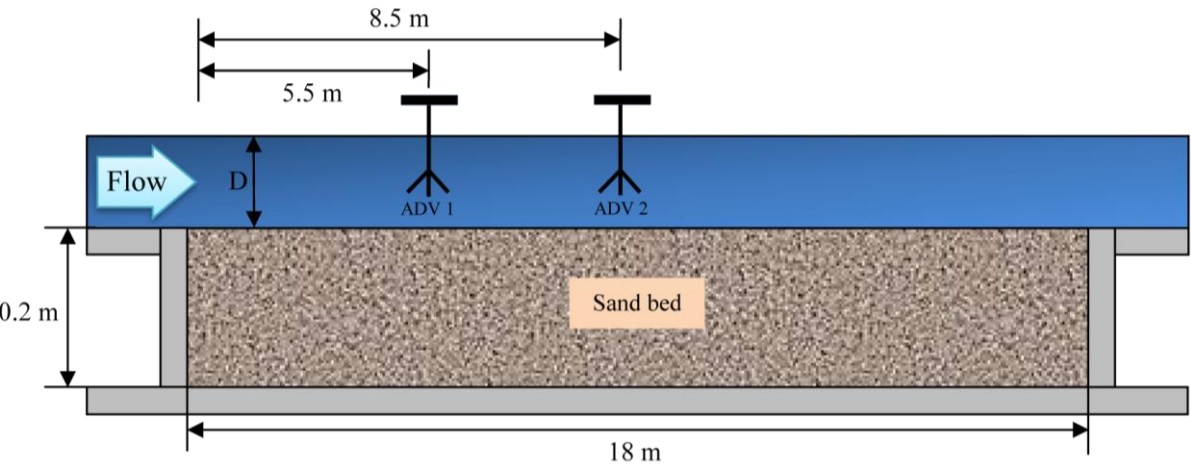

5    **Figure 2. Schematic diagram of the experimentation flume showing the key dimensions and ADV locations.**

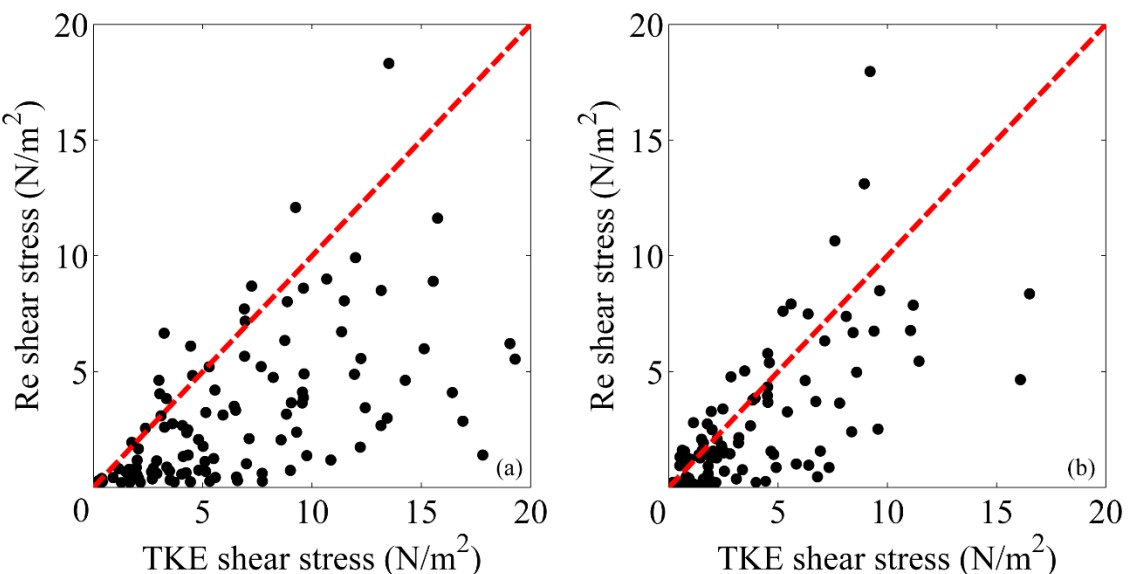

**Figure 3. Comparison of the one-second mean Reynolds and TKE shear stresses from (a) above the measured mean critical velocity ($\bar{u} > \bar{u}_{cr,\,measured}$) and (b) below the measured mean critical velocity ($\bar{u} < \bar{u}_{cr,\,measured}$) experiments with a two-minute period. The dashed red line defines the equality.**

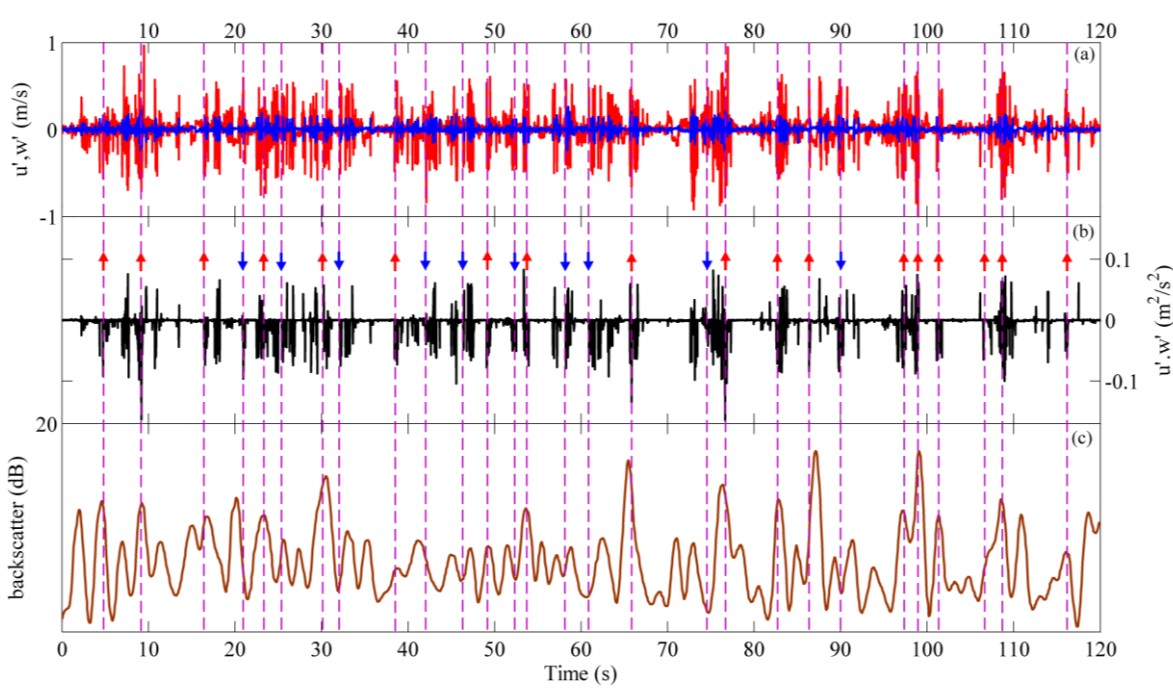

**Figure 4. Time series records from the above measured mean critical velocity experiment ($\bar{u} > \bar{u}_{cr,\,measured}$): (a) turbulent velocity (u′ - red in color, w′ - blue in color); (b) turbulent Reynolds shear stress (u′w′), showing the ejection (red up arrows) and sweep (blue down arrows) events; (c) one-second mean of the backscatter.**

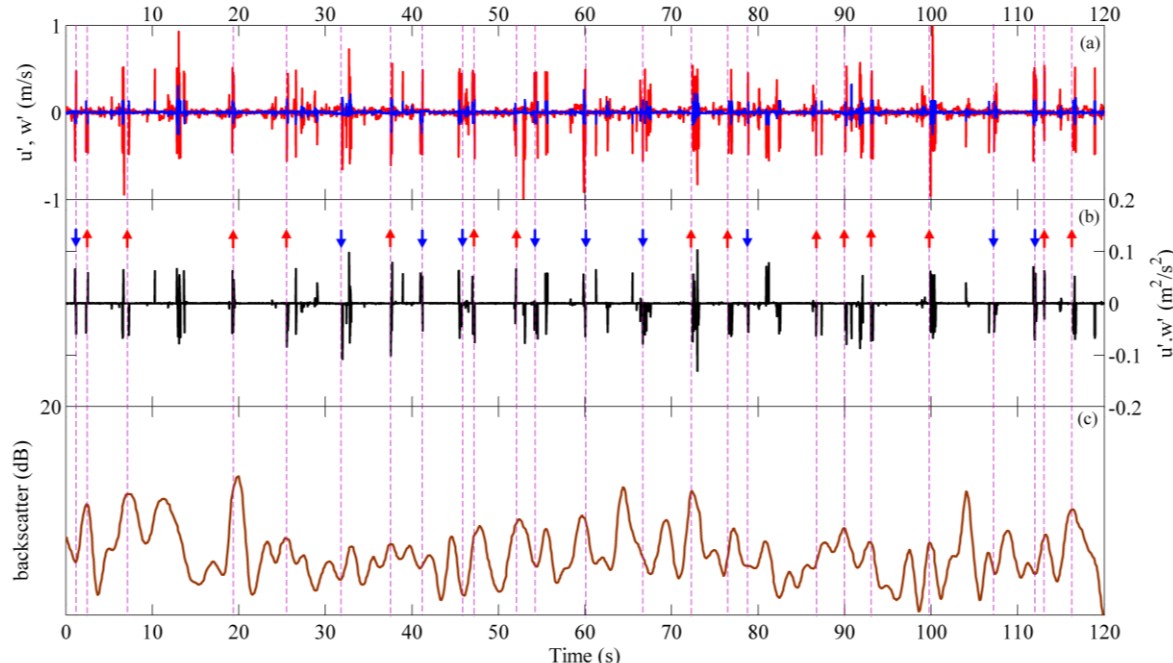

**Figure 5. Time series records from below the measured mean critical velocity experiment ($\bar{u} < \bar{u}_{cr,\ measured}$): (a) turbulent velocity (u′, w′); (b) turbulent Reynolds shear stress (u′w′), showing the ejection (red up arrows) and sweep (blue down arrows) events; (c) one-second mean of the backscatter.**

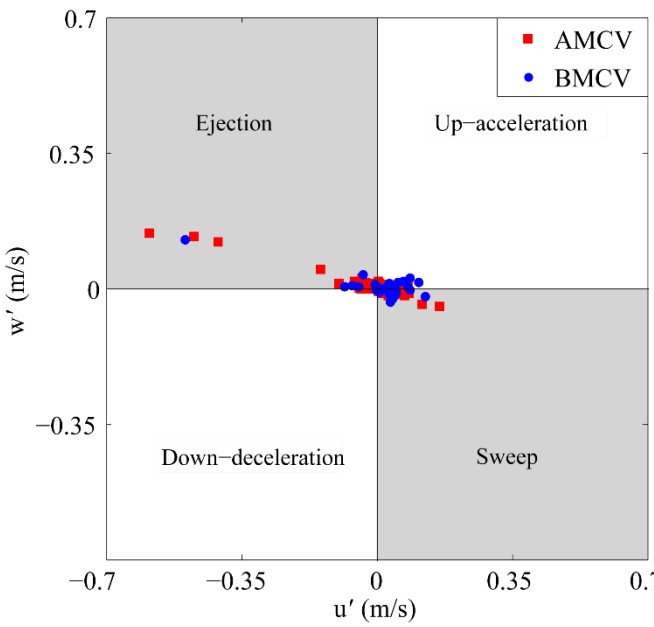

**Figure 6: Classification of bursting events in u′-w′ space identifying ejection, sweep, up-acceleration and down-deceleration events both for above and below the measured mean critical velocity conditions.**

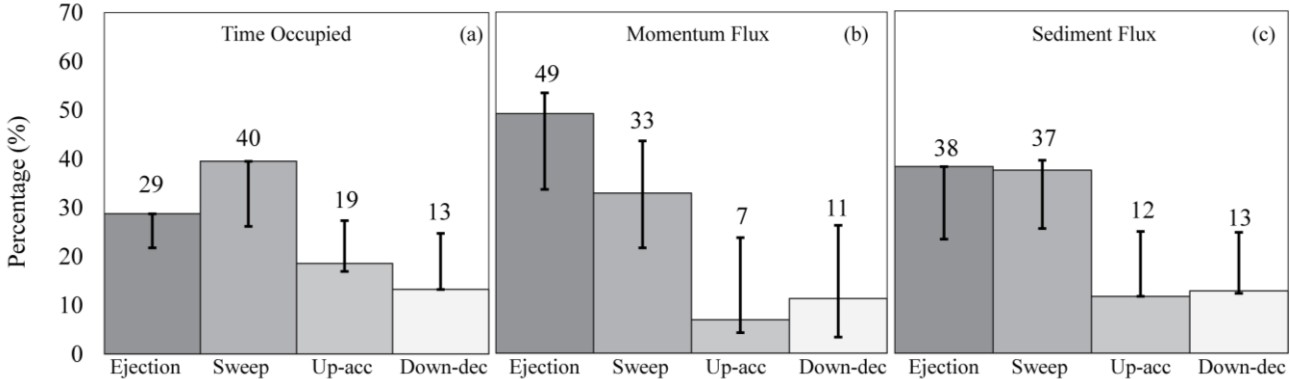

**Figure 7.** Quadrant analysis of coherent structures in above the measured mean critical velocity ranges ($\bar{u} > \bar{u}_{cr,\,measured}$) showing the (a) time occupied, (b) momentum flux (u′w′), and (c) sediment flux (c′w′). The error bars represent the maximum and minimum values of the total data.

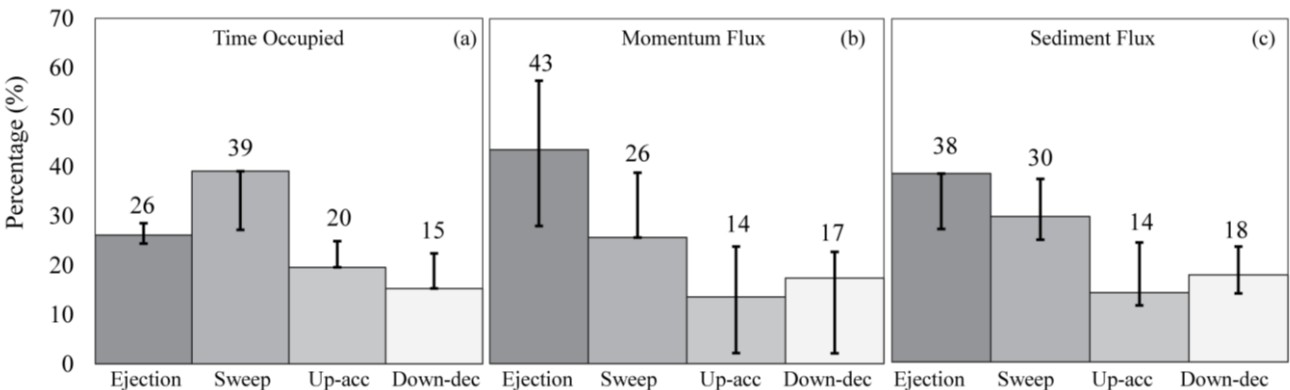

**Figure 8.** Quadrant analysis of coherent structures in below the measured mean critical velocity range ($\bar{u} < \bar{u}_{cr,\,measured}$) showing the (a) time occupied, (b) momentum flux (u′w′), and (c) sediment flux (c′w′). The error bar represents the maximum and minimum values of the total data.

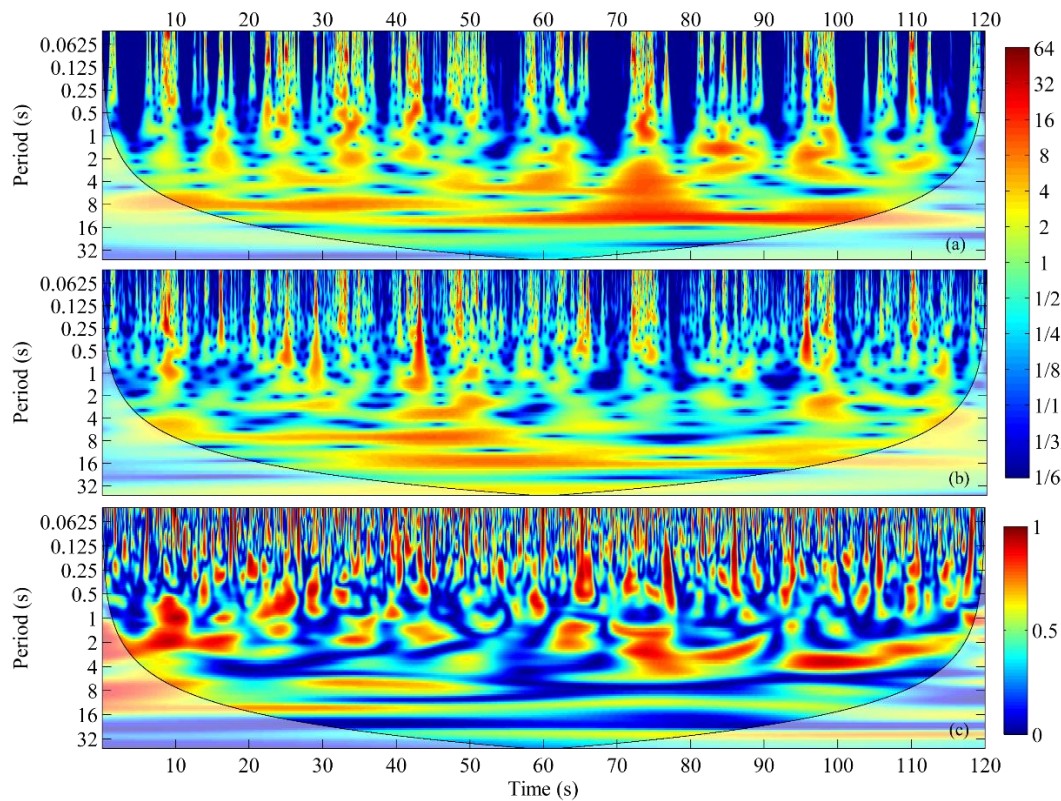

**Figure 9. Wavelet power spectra (Morlet wavelet) for above the measured mean critical velocity experiment ($\bar{u} > \bar{u}_{cr,\ measured}$) for a two-minute period showing the (a) momentum flux (u′w′), (b) sediment flux (c′w′), and (c) coherence between the momentum and sediment fluxes.**

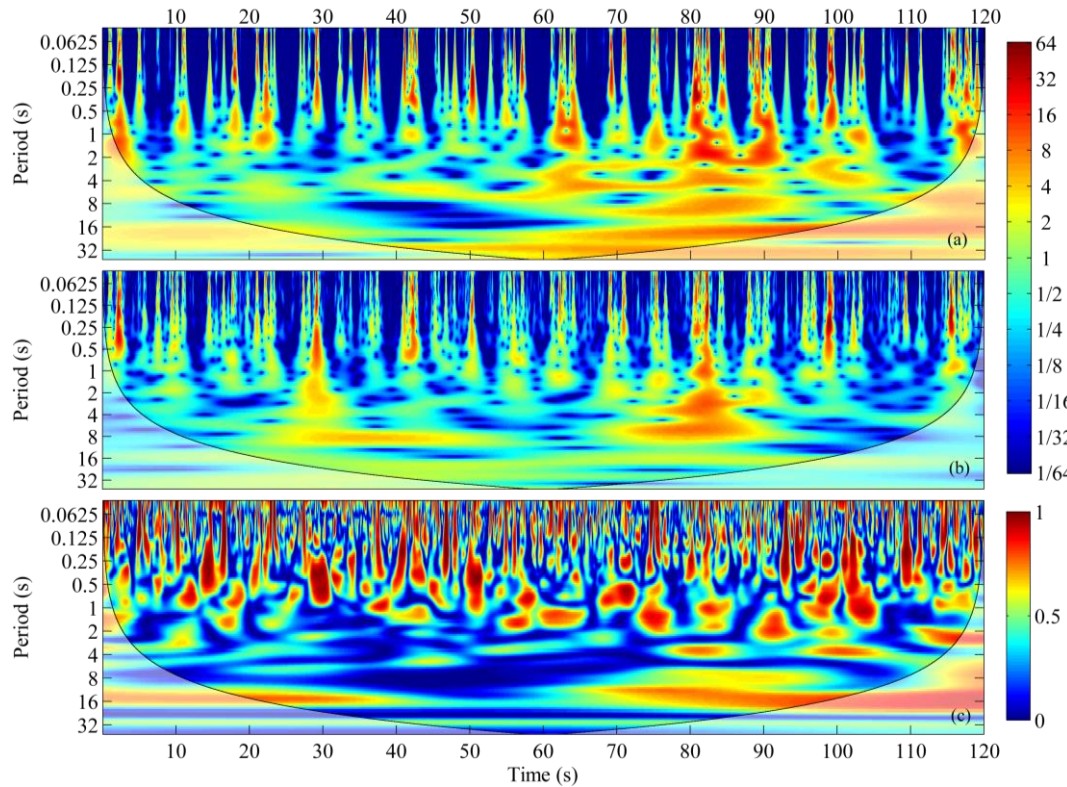

**Figure 10. Wavelet power spectra (Morlet wavelet) for below the measured mean critical velocity experiment ($\bar{u} < \bar{u}_{cr,\ measured}$) for a two-minute period showing the (a) momentum flux (u′w′), (b) sediment flux (c′w′), and (c) coherence between the momentum and sediment fluxes.**

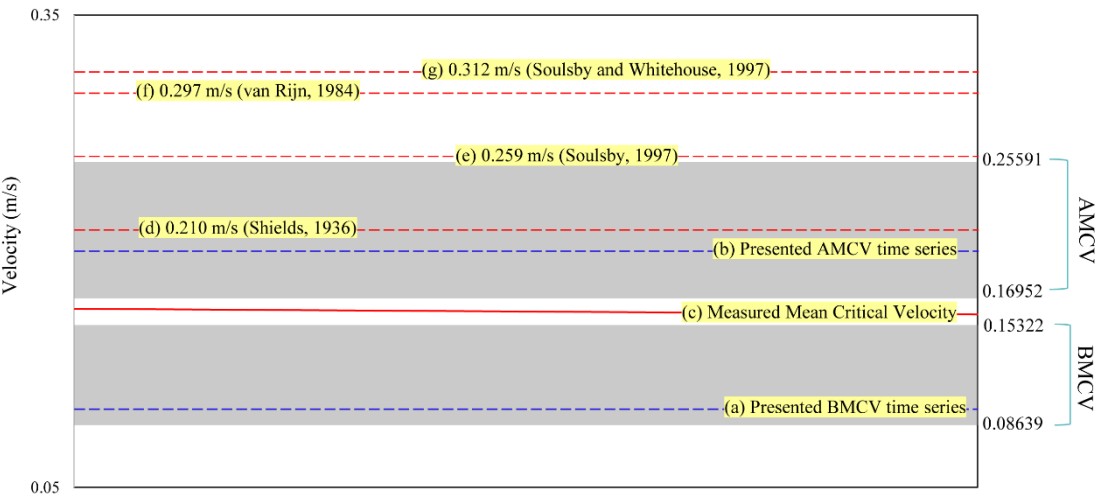

**Figure 11. Schematic diagram showing the measured and calculated mean critical velocities.**