# Peer review of "The influence of turbulent bursting on sediment resuspension under unidirectional currents"

_Earth Surface Dynamics, 2016_

## Referee Comment (RC1) · Anonymous Referee #1 · 6 Feb 2017

General Overview:

The manuscript reports on an investigation on the validity of using the critical shear velocity of sediment to define the point of sediment suspension. The research uses laboratory data provided from a separate study (Tinoco & Coco 2016) to assess the validity of a single, critical shear velocity in describing the initiation of sediment suspension under unidirectional currents. Although I support the approach, I do not think the manuscript, at present, contributes any new ideas to the subject. The manuscript is missing references to many key authors and papers on the subject, some notable exceptions include (Garcia et al., 1996; Schmeeckle & Nelson 2003; Diplas et al., 2008; Valyrakis et al., 2013; Keylock et al., 2014); for initiation of motion; Shugar et al.,

(2010) for suspended sediment and wavelets; and (Falco 1991; Adrian 2007) for the structure of wall bounded flows, burst and sweeps and coherent flow structures. The lack background knowledge of the subject is demonstrated when reading the discussion section, which does not do a good job in bringing the results into the wider context of the subject.

In light of the fact that the first author is presently conducting their PhD. My decision is that the manuscript should be reconsidered after major revisions, rather than reject.

The manuscript does well at:

The description of the experimental method is detailed enough to be reproducible. Grammar and references and up to standard.

Suggestions for improvement:

For the manuscript to progress any further, the authors need to make an attempt to quantify the effect of turbulent bursting on the initiation of sediment suspension. At present the manuscript only calls into question the use of a critical shear velocity, which was originally done by (Grass 1970) (if not earlier) and an attempt should be made to define an adjustment to or alternative to using a single value for the critical shear velocity, or even for using it. A positive outcome, or an indication of where future research should lead, would make the manuscript much stronger, such as the cited paper (Tinoco & Coco 2016)

The introduction needs to be re-written, with the last paragraph, which contains much of the importance of the research integrated into the first paragraph. The fundamental physics of sediment suspension needs to be detailed, ideally with the governing equations, so a description of the physical processes operating and under investigation can be described and mapped onto the experimental methods. A paper about critical shear velocity really should contain the equations used to calculate the critical shear velocity.

At present the discussion is very weak and does not develop upon the results other

than qualitative explanations. The discussion needs to bring in the wider literature so the implications of the results are clearer. As a general comment, throughout the discussion, vague and qualitative terms like "Considerably", "reasonably", and "close to" need to be replaced with quantitative values and specific reference to the results of the work. It is my opinion that the manuscript cannot progress further unless this section has been re-worked.

What is the difference (if any) between sediment suspension and re-suspension?

As the authors are using a critical shear velocity which was measured with the instruments that are used in the present study. So, why does the measured critical shear velocity of the sediment seem such a bad predictor of sediment suspension? The authors need to define in this manuscript exactly how their value of critical shear velocity was produced because at present this does not make any sense, other than their measured critical shear velocity is wrong.

Line by line comments

Page 1 Line 13: Is the manuscript about incipient motion or suspension? Page 1 Line 30: lower than what? Equations would be useful here and help define form and viscous drag Page 2 Line 1: define the Reynolds number, and particle Reynolds number with the equations and directly relate to page 1 line 30 Page 2 line 3: It might be obvious, but state which critical value Page 2 Line 5: Where does the limited applicability come from? This is a very important point and needs to be explained properly, in fact the next sentence contradicts this. Page 2 Line 10: You've half made the point, this needs detail. What did Dey 2011 do? How does your work follow on from those advances? Page 2 Line 11 to 20: you need to think about what point you are trying to make with this paragraph and how does it fit in with the rest of the introduction. This is background info that should come before you talk about more recent developments. Page 4 line 19: how need to show where the measured value of critical suspension velocity sits on the shield curve. What are the Rouse number for the experiments? I estimate ACV

= 3.05 and BCV to be 6.36 using a fall velocity calculated from (Ferguson & Church 2004). How are you defining suspension? It looks like you're measuring the initiation of motion rather than suspension here? Page 7-8, the paragraph on wavelets is full of incredibly vague terms and needs a complete re-write. Qualifiers such as: "Fast, slow, large, small gradually, sporadically, longer, shorter, weakening" needs removing. Make the results quantifiable and cite the figures and the data. The lists of identified events is somewhat useful and is a good attempt at quantification but it is not easy to use as a reader. Maybe a table of the data, with sweep and ejections in adjacent columns could be easier to read? Maybe colour the table cells by the value of the cell to make it easier to see the relationships. Page 7 Line 19: the sentence starting here doesn't make sense. Page 7 Line 24: multiscale and fine scale, large scale. These are very qualitative measurements! Page 7 Line 26: "highly energy turbulent events" what do you mean by this? How high is high? Page 7 Line 28: "dominant direction of flow near the bed" is that direction u, v or w? Very vague Page 8 Line 7: "common features were noticed". . . no! cite the figures, maybe identify these common features on the figures.

References Adrian, R.J., 2007. Hairpin vortex organization in wall turbulence. Physics of Fluids, 19(4), p.41301. Diplas, P. et al., 2008. The Role of Impulse on the Under Turbulent Flow Conditions. Science, 322, pp.717–720. Falco, R.E., 1991. A coherent structure model of the turbulent boundary layer and its ability to predict Reynolds number dependence. Philosophical Transactions: Physical Sciences and Enineering, 336(1641), pp.103–129. Ferguson, R.I. & Church, M., 2004. A Simple Universal Equation for Grain Settling Velocity. Journal of Sedimentary Research, 74(6), pp.933–937. Garcia, M.H., Nino, Y. & Lopez, F., 1996. Laboratory observations of Particle Entrainment into suspension by turbulent bursting. In P. J. Ashworth et al., eds. Coherent flow Structures in Open Channel Flows. Grass, J., 1970. Initial instability of fine bed sand. Journal of Hydraulic Division. American Society of Civil Engeiners, 93, pp.619–631. Keylock, C.J., Lane, S.N. & Richards, K.S., 2014. Quadrant/octant sequencing and the role of coherent structures in bed load sediment entrainment. Journal of Geophysical Research: Earth Surface, 119(2), pp.264–286. Schmeeckle, M.W. & Nelson,

J.M., 2003. Direct numerical simulation of bedload transport using a local , dynamic boundary condition. Sedimentology., 50, pp.279–301. Shugar, D.H. et al., 2010. On the relationship between flow and suspended sediment transport over the crest of a sand dune, RiÂ■o Parana, Argentina. Sedimentology, 57(1), pp.252–272. Tinoco, R.O. & Coco, G., 2016. A laboratory study on sediment resuspension within arrays of rigid cylinders. Advances in Water Resources, 92, pp.1–9. Valyrakis, M., Diplas, P. & Dancey, C.L., 2013. Entrainment of coarse particles in turbulent flows: An energy approach. Journal of Geophysical Research: Earth Surface, 118(1), pp.42–53.
* * *

---

## Referee Comment (RC2) · Anonymous Referee #2 · 20 Mar 2017

The manuscript by Salim et al. describes a series of experiments of open channel flow over flat surface of sediments to study the definition of the threshold velocity necessary for suspension of sediments. The authors chose two mean flow velocities, one above and another below the critical velocity and examined sediment flux correlation with momentum flux as well as turbulent ejection and sweep events. I found the manuscript well written and appropriate for publication after revision. However, several issues, most minor, must be addressed regarding this manuscript before this effort becomes an acceptable publication in my point of view.

1. Some information regarding the open channel flow is missing that would be very beneficial for fluid mechanics when they require to compare the features of the flow: Reynolds number based on the wall shear velocity, particle Reynolds number based on the wall shear velocity. Other than the bulk Reynolds number, another relevant non-dimensional number is the particle Reynolds number with an appropriate velocity scale that here should be the shear velocity. Can authors compute/estimate the shear velocity using momentum balance?

2. The last sentence of Introduction needs more elaboration. It is a jump to a literature without explaining it: The critical bed shear stress can also be excluded when computing the bedload grain velocity (Cheng and Emadzadeh, 2014).

3. A few literatures are missing in the paper:
   a. Robinson, S.K., 1991. Coherent motions in the turbulent boundary layer. *Annual Review of Fluid Mechanics*, *23*(1), pp.601-639.
   b. Bagnold, R.A., 1956. The flow of cohesionless grains in fluids. *Philosophical Transactions of the Royal Society of London A: Mathematical, Physical and Engineering Sciences*, *249*(964), pp.235-297.
   c. van Rijn, L.C., 2013. Simple general formulae for sand transport in rivers, estuaries and coastal waters. *Retrieved from www. leovanrijn-sediment. com*.

4. Since the experiments are occurred over flat surface, I do suggest to mention this explicitly in the title or Abstract; "unidirectional currents over flat bed"

5. Discussion of Figure 3 is vague! How do authors conclude that "Sufficient shear stress was produced to generate sediment resuspension"?

6. What do authors suggest in order to improve the current representation of the threshold? They should mention a variable that may correlate better to these phenomena than shear stress.

7. Line 26, page 7, change "energy" to "energetic".

---

## Editor Comment (EC1) · D.R Parsons (Editor) · 20 Mar 2017

Dear Authors, There are two reviews now in place for your manuscript....both are generally supportive but do have some substantive concerns in places. Please consult these reviews, amend the manuscript in light of the comments and provide a point-by-point response to each comment. Many thanks Daniel R. Parsons
* * *

---

## Author Comment (AC1) · 24 Apr 2017

The authors are thankful for the Reviewer's comments and suggestions to improve the manuscript. Significant parts of the manuscript have been modified and extended to address the main concerns, in particular the novelty of the proposed analysis and how it fits within and distinguishes from existing literature on similar subjects.

We have structured the response to Reviewer 1 in two parts:

A) addressing the main concerns and implementation of the Reviewer 1 suggestions for improvement;
B) our response to the line-by-line comments of Reviewer 1

**Response to Reviewer 1.**

**A) Major modifications**

The reviewer states:

*" I do not think the manuscript, at present, contributes any new ideas to the subject. The manuscript is missing references to many key authors and papers on the subject, some notable exceptions include (Garcia et al., 1996; Schmeeckle & Nelson 2003; Diplas et al., 2008; Valyrakis et al., 2013; Keylock et al., 2014); for initiation of motion; Shugar et al., (2010) for suspended sediment and wavelets; and (Falco 1991; Adrian 2007) for the structure of wall bounded flows, burst and sweeps and coherent flow structures. The lack background knowledge of the subject is demonstrated when reading the discussion section, which does not do a good job in bringing the results into the wider context of the subject."*

After a careful review of the suggested literature, the manuscript has been modified to account for the missing references and place our work within a wider context, explicitly stating the differences between our analyses and previous ones, and why we believe this manuscript brings a new approach to the subject.

We have added additional paragraphs in the Introduction [page 4, line 20 and page 5, line 7]

The Discussion section has also been modified to address the reviewer remarks (page 13, line 2.

*Reviewer suggestions for improvement, including all clarifications and modifications to the original manuscript are addressed below.*

**A1.** *For the manuscript to progress any further, the authors need to make an attempt to quantify the effect of turbulent bursting on the initiation of sediment suspension. At present the manuscript only calls into question the use of a critical shear velocity, which was originally done by (Grass 1970) (if not earlier) and an attempt should be made to define an adjustment to or alternative to using a single value for the critical shear velocity, or even for using it. A positive outcome, or an indication of where future research should lead, would make the manuscript much stronger, such as the cited paper (Tinoco & Coco 2016).*

The aim of the paper, rather than developing a better transport equation was to highlight the importance of instantaneous events on sediment re-suspension, that were not considered when using the classical shields diagram approach that uses a mean velocity concept. While Grass (1970), Lavelle and Mofgeld (1987) and other comprehensive surveys discussed by Miller et al. (1977), Buffington and Montgomery (1997), Paphitis (2001), and, Dey and Papanicolaou (2008) pointed out that turbulence rather than mean shear stress has direct contributions to

sediment motion. We compliment these studies by using advanced instrumentation and high resolution laboratory data to document turbulence ('bursting') events and their relationship to sediment resuspension. As currently the turbulence events cannot be predicted it was not possible to develop a predictive framework from the experimental data.

**A2. The introduction needs to be re-written, with the last paragraph, which contains much of the importance of the research integrated into the first paragraph. The fundamental physics of sediment suspension needs to be detailed, ideally with the governing equations, so a description of the physical processes operating and under investigation can be described and mapped onto the experimental methods. A paper about critical shear velocity really should contain the equations used to calculate the critical shear velocity.**

The introduction has been rewritten, including a theoretical framework for particles in resuspension to facilitate understanding of the analysis, and calculation of critical shear stress has been added (modified paragraph at page 1, line 25).

**A3. At present the discussion is very weak and does not develop upon the results other than qualitative explanations. The discussion needs to bring in the wider literature so the implications of the results are clearer. As a general comment, throughout the discussion, vague and qualitative terms like "Considerably", "reasonably", and "close to" need to be replaced with quantitative values and specific reference to the results of the work. It is my opinion that the manuscript cannot progress further unless this section has been re-worked.**

All qualitative assertions have been replaced by quantitative references to our data (page 13, line 2).

**A4. What is the difference (if any) between sediment suspension and re-suspension?**

For this manuscript we use the term 'resuspension' for particles which were initially on the bed and at some point lifted from the bed into the water column, rather than particles permanently in suspension (washload).

*"ejections are associated with entrainment of sediment particles into water column, while sweeps are effective at transporting bedload (Cao, 1997; Dyer and Soulsby, 1988; Heathershaw, 1979; Keylock, 2007; Soulsby, 1983; Yuan et al., 2009) in Kassem et al., 2015"*

*\*Kassem, H., Thompson, C. E. L., Amos, C. L. and Townend, I. H.: Wave-induced coherent turbulence structures and sediment resuspension in the nearshore of a prototype-scale sandy barrier beach. Continental Shelf Research, 109, 78-94, 2015.*

**A5. As the authors are using a critical shear velocity which was measured with the instruments that are used in the present study. So, why does the measured critical shear velocity of the sediment seem such a bad predictor of sediment suspension? The authors need to define in this manuscript exactly how their value of critical shear velocity was produced because at present this does not make any sense, other than their measured critical shear velocity is wrong.**

This manuscript reports on an investigation on the validity of using the mean critical shear velocity of sediment to define initiation of sediment transport.

Now the question is if we have 'measured' critical velocity ($\bar{u}_{cr}$ measured = 0.163 m/s) then we should not see any sediment resuspension in our BMCV time series. But it is still evidencing resuspension. It should be highlighted that the 'measured' critical velocity is a mean value (see figure below) and we demonstrate that sediment resuspension can occur below this mean value.

To answer that, we can look into the Tinoco and Coco (2013) paper where it says:

………………The determination of thresholds from experimental data can be a topic of discussion by itself. In our case, the thresholds were determined by finding the velocity at which the instrument starts recording concentrations higher than the background levels. The predicted values (Table 4) are considerably higher than the ones observed during the experiments, not only in the populated cases, where the cylinders clearly enhance sediment resuspension, but also in the smooth sand bed case, which can be explained either by irregularities in the flat initial conditions of the sand bed, or by the effect of the intrusive instrumentation deployed (steel rods protruding through the sand to hold ADVs and OBSs). The height of the measurements (5 cm above the bed) must also be considered, since the physical dimensions of the instruments prevent us from getting closer to the bed, where material could be already in suspension at lower elevations before the OBSs are able to record it………………

So, the threshold was considered when OBS started recording the mean concentration higher than the mean background. That means critical velocity was taken as the point of shifting the 'mean' concentration from one point to the higher point. Please look at the following sketch for better clarification:

[Figure]

As the change is in the mean, the fluctuating part was ignored in the 'measured critical velocity'. This is why in our BMCV condition we still found the sediments to resuspended which are related to those fluctuations. From the Tinoco & Coco 2013, 2016 experiments, a critical resuspension velocity was found as the nearbed (5 cm above the bed) mean velocity at which a turbidity sensor (optical backscatter sensor, also located at 5 cm above the bed) started measuring an increase with respect to the background concentration.

Therefore, the aim of this paper is to highlight that all our methods of measuring/ calculating the critical velocity is overlooking that the sediment transport process is an event based system where the averaging approach does not hold. Future transport equations should consider the fluctuations rather the mean.

**B) Line by line comments:**

*B1. Page 1 Line 13: Is the manuscript about incipient motion or suspension?*

As mentioned in comment number (A4) above, this manuscript is not about 'particles permanently in suspension (washload)' instead it is about the particles which are initially laying on the bed and at some point, lifted from the bed into the water column which we defined as 'resuspension'. However, we fully agree with the reviewer that it has created some confusion in the paper. Therefore, we rewrote the sentence at page 1, line 13. Now reads:

"…………… to examine the role of turbulence on sediment resuspension"

*B2. Page 1 Line 30: lower than what? Equations would be useful here and help define form and viscous drag*

*B3. Page 2 Line 1: define the Reynolds number, and particle Reynolds number with the equations and directly relate to page 1 line 30*

We fully agree with the reviewer that relevant equations are helpful here to define form drag, viscous drag, Reynolds number and particle Reynolds number. Therefore, we revised, and directly related the page 1 line 30 and page 2 line 1. Now reads:

"At velocities lower than the threshold, shear stress represented the viscous drag imparted by the moving fluid to the bed particles whereas at velocities higher than the critical, it was related to the pressure differential between the upstream and downstream sides of the particle. Shields also defined the non-dimensional critical shear stress, $\theta_{cr}$, as a function of the boundary Reynolds number, $Re_p$, defined as:

$$\theta_{cr} = \tau_o/(\rho_s - \rho)gd_s \tag{1}$$
$$Re_p = u_* d_s/\upsilon \tag{2}$$

where, $\tau_o$ is the critical bottom shear velocity, $\rho_s$ and $\rho$ are the sediment and fluid densities, $g$ is the acceleration due to gravity, $d_s$ is the particle diameter, $u_* = \sqrt{\tau_o/\rho}$ is the critical shear velocity and $\upsilon$ is the kinematic viscosity of the fluid."

*B4. Page 2 line 3: It might be obvious, but state which critical value*

We agree with reviewer's concern here and revised it at page 2, line 13. Now reads:

"Such criterion (commonly used via a Shields diagram, e.g., Kennedy, 1995; Buffington, 1999, Paphitis, 2001) states that sediment is entrained once bed shear stress exceeds the Shields mean critical value."

*B5. Page 2 Line 5: Where does the limited applicability come from? This is a very important point and needs to be explained properly, in fact the next sentence contradicts this.*

We fully agree with reviewer that the stated sentence highlighting previous studies with limited applicability was explained inappropriately. Moreover it contradicted the next sentence. As this statement is very important therefore, we carefully addressed it at page 2, line 15. Now reads:

"The impact of turbulence, however, was traditionally represented only by a mean quantity such as Reynolds shear stress (e.g. widely used bedload and suspended load formulations presented in van Rijn, 2013). Further attempts to characterise sediment entrainment advocated that it solely depended on fluid lifting force, with near bed sediment being entrained due to instantaneous near bed vertical velocity (Einstein, 1950; Velikanov, 1955; Yalin, 1963; Ling, 1995). In contrast, Bagnold (1956) hypothesised that particles remain in suspension as long as the turbulent eddies have dominant vertical velocity components, which would scale with the flow shear velocity, that exceed the particle settling velocity. It implies that to establish a dynamic equilibrium of sediment exchange, the flow must continuously pick up the sediment at the same rate with an upward velocity equalling terminal fall velocity."

*B6. Page 2 Line 10: You've half made the point, this needs detail. What did Dey, 2011 do? How does your work follow on from those advances?*

*B7. Page 2 Line 11 to 20: you need to think about what point you are trying to make with this paragraph and how does it fit in with the rest of the introduction. This is background info that should come before you talk about more recent developments.*

We agree with the reviewer's comments here, merged and rewrote the paragraphs. Please read at page 2, line 24.

*B8. Page 4 line 19: how need to show where the measured value of critical suspension velocity sits on the shield curve. What are the Rouse number for the experiments? I estimate ACV = 3.05 and BCV to be 6.36 using a fall velocity calculated from (Ferguson & Church 2004). How are you defining suspension? It looks like you're measuring the initiation of motion rather than suspension here?*

We agree with the reviewer that the lack of providing a clear definition created the confusion in the manuscript. However, after carefully revising the paper as stated above in the section (A4), this paper is related to sediment resuspension rather suspension or incipient motion. Therefore, the concern of the Rouse number is beyond the scope of this paper. Anyway, the values have been calculated and included for the sake of completeness, as seen in page 6, line 12. Now reads:

"Data from the near-bed ADV 1 is presented in this manuscript where the mean flow speeds, ū, varied from 0.087 to 0.256 m/s, covering a range of boundary Reynolds number, $Re_p$={342-1004}; flow Reynolds number, $Re_D$=(ūD/υ)={$1.4×10^4$-$4.1×10^4$} and Rouse number, P= $w_s/ku_*$= {2.89-8.14} where $u_*$ was calculated using the bed shear stress computed with Eq.4 at z=5 cm, ū was mean velocity, $k_s$ was the von Kármán constant (assuming as 0.41) and $w_s$ was particle fall velocity calculated from Dietrich (1982)."

*B9. Page 7-8, the paragraph on wavelets is full of incredibly vague terms and needs a complete re-write. Qualifiers such as: "Fast, slow, large, small gradually, sporadically, longer, shorter, weakening" needs removing. Make the results quantifiable and cite the figures and the data. The lists of identified events is somewhat useful and is a good attempt*

*at quantification but it is not easy to use as a reader. Maybe a table of the data, with sweep and ejections in adjacent columns could be easier to read? Maybe colour the table cells by the value of the cell to make it easier to see the relationships.*

We agree with the review's concern and completely revised the paragraph related to wavelets removing vague terms. Please read at page 11, line 20 and Table 2.

*B10. Page 7 Line 19: the sentence starting here doesn't make sense.*

We agree with reviewer and removed the sentence.

*B11. Page 7 Line 24: multiscale and fine scale, large scale. These are very qualitative measurements!*

We agree with the reviewer, included quantitative measurements. Please read at page 11, line 26.

*B12. Page 7 Line 26: "highly energy turbulent events" what do you mean by this? How high is high?*

We agree with the reviewer, corrected it at page 11, line 31. Now reads:

"…….high period (warmer colour >0.5s) associated with the mean flow properties for both momentum flux and sediment flux."

*B13. Page 7 Line 28: "dominant direction of flow near the bed" is that direction u, v or w? Very vague*

We agree with the reviewer, corrected it. Please read at page 12, line 3. Now reads:

"…..in the dominant streamwise-vertical plane of the flow near the bed,….."

*B14. Page 8 Line 7: "common features were noticed". . . no! cite the figures, maybe identify these common features on the figures.*

We agree with the reviewer and revised this section with the inclusion of a new table. Please read at page 12, line 15 and Table 2.

---

## Author Comment (AC2) · 24 Apr 2017

The authors are thankful for the Reviewer's comments and suggestions to improve the manuscript. Significant parts of the manuscript have been modified and extended to address the main concerns, in particular the novelty of the proposed analysis and how it fits within and distinguishes from existing literature on similar subjects.

We have structured the response to Reviewer 2 in two parts:

C) addressing main concerns of the Reviewer 2, followed by
D) our response to the line-by-line comments of Reviewer 2.

**Response to Reviewer 2.**

**C) Major modifications**

The reviewer rightfully states:

*C1. Some information regarding the open channel flow is missing that would be very beneficial for fluid mechanics when they require to compare the features of the flow: Reynolds number based on the wall shear velocity, particle Reynolds number based on the wall shear velocity. Other than the bulk Reynolds number, another relevant non-dimensional number is the particle Reynolds number with an appropriate velocity scale that here should be the shear velocity. Can authors compute/estimate the shear velocity using momentum balance?*

We agree with the reviewer, included it (page 2, line 3).

*C2. The last sentence of Introduction needs more elaboration. It is a jump to a literature without explaining it: The critical bed shear stress can also be excluded when computing the bedload grain velocity (Cheng and Emadzadeh, 2014).*

We revised the last paragraph of the Introduction. Now reads:

"Conclusions reached by these authors agree that a single mean value of shear stress is not an accurate estimate for sediment transport, and further consideration must be given to instantaneous turbulent parameters for a better characterisation of flow-sediment interactions."

For details please read the full paragraph stated at page 2, line 24.

*C3. A few literatures are missing in the paper:*

*a. Robinson, S.K., 1991. Coherent motions in the turbulent boundary layer. Annual Review of Fluid Mechanics, 23(1), pp.601-639.*

*b. Bagnold, R.A., 1956. The flow of cohesionless grains in fluids. Philosophical Transactions of the Royal Society of London A: Mathematical, Physical and Engineering Sciences, 249(964), pp.235- 297.*

*c. van Rijn, L.C., 2013. Simple general formulae for sand transport in rivers, estuaries and coastal waters. Retrieved from www. leovanrijnsediment. com.*

Included (page 2, line 19; page 3, line 14; page 2, line 16).

*C4. Since the experiments are occurred over flat surface, I do suggest to mention this explicitly in the title or Abstract; "unidirectional currents over flat bed"*

Mentioned in the Abstract (page 1, line 12) and in the Introduction (page 5, line 16).

*C5. Discussion of Figure 3 is vague! How do authors conclude that "Sufficient shear stress was produced to generate sediment resuspension"?*

We have modified the discussion to clarify our observations (page 10, line 2). It now reads:

"The scatterplots of the Reynolds and TKE bottom shear stresses for the AMCV and BMCV runs (Figs. 3a and 3b) showed that higher bed shear stress (i.e. values >5 N/m$^2$ of TKE and Re shear stress estimations of both AMCV and BMCV runs) was produced to generate sediment resuspension (as evidenced with backscatter intensity on Figs. 4c and 5c). Such comparison of the TKE and Re shear stress methods also suggested the presence of coherent flow structures in the turbulent flow which created highly localised and persistent variability near the bed, hence affecting the bed shear stress."

*C6. What do authors suggest in order to improve the current representation of the threshold? They should mention a variable that may correlate better to these phenomena than shear stress.*

We suggested the following in the conclusion section of the revised paper:

"Our statistical assessment suggests that the existing definition of threshold can be improved by incorporating turbulent effects for a more accurate description of the processes involved which will result in better predictions of sediment transport. The results of this study are instrumental in resolving an important research question: how best to incorporate the turbulent bursting events into a theoretical model describing the sediment entrainment process? The analysis detailed herein on identification of bursting events and their contribution toward the near-bed Reynolds shear stress production governing sediment motion provide new avenues to answer such question, incorporating the use of wavelet analysis on time series of acoustic backscatter or signal intensity readily available from commonly used acoustic velocimetry instruments (ADVs and ADCPs) as a powerful tool for investigating such processes. The fact that a similar methodology can be applied to existing field and laboratory datasets that focused on velocity but collected an indicator of signal backscatter as part of the data record, further highlights its potential in future research to elucidate a more complete understanding of the interactions between flow and sediment transport over complex topography."

Please also refer to Response to Reviewer 1 (section A1) highlighting the objective of this paper.

**D) Line by line comments:**

*D1. Line 26, page 7, change "energy" to "energetic".*

Corrected.

---

## Editor Comment (EC2) · D.R Parsons (Editor) · 31 May 2017

The authors have done a good job at addressing the main points raised by the reviewers and have made some significant edits to the manuscript in response to the suggestions of the reviewers. There are however one or two areas where I would like the authors to go a little further. In the rebuttal the following sentence is put forward:

"The aim of the paper, rather than developing a better transport equation was to highlight the importance of instantaneous events on sediment re-suspension, that were not considered when using the classical shields diagram approach that uses a mean velocity concept." and "We compliment these studies by using advanced instrumentation and high resolution laboratory data to document turbulence ('bursting') events and their

relationship to sediment resuspension. As currently the turbulence events cannot be predicted it was not possible to develop a predictive framework from the experimental data."

......the first part needs to be evolved into the introduction to delineate the study more fully for the readers....some of this is there, but I think it could be made clearer by some minor edits. Secondly, the final sentence is not actually true (i.e. DNS modelling) so this needs more careful language in the discussion of the paper - highlighting the importance and the direction of travel and how the work in this paper meshes with other approaches.

Subject to these amendments I recommend publication, Daniel R. Parsons

---

## Author Comment (AC4) · 3 Jun 2017

The authors are thankful for the Editor's comments and suggestions to improve the manuscript. The manuscript has now been modified and extended to address the main concerns raised by the Editor. In particular, we are better specifying the goal of the manuscript to avoid misinterpretations, and framing our work with respect to other approaches (DNS), addressing potential future directions where the proposed methodology and DNS can provide valuable feedback to better characterise complex flow scenarios. The specific modifications are listed below.

**Response to Editor's comments:**

*"The aim of the paper, rather than developing a better transport equation was to highlight the importance of instantaneous events on sediment re-suspension, that were not considered when using the classical shields diagram approach that uses a mean velocity concept."*

*….the first part needs to be evolved into the introduction to define the study more fully for the readers....some of this is there, but I think it could be made clearer by some minor edits.*

We agree with the editor, corrected it at page 5, line 9. Now reads:

"The aim of the paper, rather than developing a better transport equation, is to highlight the importance of instantaneous events on sediment resuspension, which were not considered when using the classical Shields diagram approach that uses a mean velocity concept."

*"We compliment these studies by using advanced instrumentation and high resolution laboratory data to document turbulence ('bursting') events and their relationship to sediment resuspension. As currently the turbulence events cannot be predicted it was not possible to develop a predictive framework from the experimental data."*

*Secondly, the final sentence is not actually true (i.e. DNS modelling) so this needs more careful language in the discussion of the paper - highlighting the importance and the direction of travel and how the work in this paper meshes with other approaches.*

We agree with the comment, and have modified relevant texts accordingly.

Page 14, line 12, now reads:

"In contrast, direct numerical simulations (DNS) provide a new tool for examining turbulent structure of the flow (e.g. Mathis et al. 2013). However, further development is required to apply the DNS approach to intermittent turbulent bursting events both in fluvial and geophysical flows (Venditti et al., 2013). For example, Mathis et al. (2014) estimated bed shear-stress using conventional methods and DNS modelling approach and reported a large disparity between the two methods where an order of magnitude difference between the levels of energy spectra observed. While DNS has the potential to develop methodologies for the prediction of bursting events and associated sediment resuspension mechanisms, its application on large-scale, complex flows still remains limited (e.g. Schmeeckle and Nelson, 2003). Experimental investigations such as the one developed herein, will allow to use new and existing data from acoustic velocimetry sensors to further identify and characterise such turbulent events. Direct observations of bursting events will in turn better inform DNS methods to better account for flow interactions."

Page 15, line 25, now reads:

"Such observations presented in this paper are also necessary to clarify our view of turbulent coherent structures in resuspending sediments both in low and high Reynolds-number flows while leading widespread application of DNS."

---

## Author Comment (AC5) · 3 Jun 2017

The comment was uploaded in the form of a supplement:
http://www.earth-surf-dynam-discuss.net/esurf-2016-60/esurf-2016-60-AC5-supplement.pdf